# A Comprehensive Review on Multifaceted Mechanisms Involved in the Development of Breast Cancer Following Adverse Childhood Experiences (ACEs)

**DOI:** 10.3390/ijerph191912615

**Published:** 2022-10-02

**Authors:** Olimpia Pino, Rosalinda Trevino Cadena, Diana Poli

**Affiliations:** 1Department of Medicine and Surgery, University of Parma, 43125 Parma, Italy; 2INAIL Research, Department of Occupational and Environmental Medicine, Epidemiology and Hygiene Via Fontana Candida 1, Monte Porzio Catone, 00078 Rome, Italy

**Keywords:** Adverse Childhood Experiences, breast cancer, stress, childhood trauma, biomarkers, depression, anxiety

## Abstract

Background and aim of the work. Adverse Childhood Experiences (ACEs) may give rise to harmful effects on health throughout life. Epigenetic changes explain how preexisting risk factors may contribute to produce altered biological responses and cancer risk. The main aim of the review is to summarize studies examining the means in which Adverse Childhood Experiences (ACEs) can modulate individual vulnerability to breast cancer (BC) development through multifaceted mechanisms. Methods. Studies selection, data extraction, and assessments agreed to PRISMA criteria. We included original research with clinical samples following BC interventions, investigating potential mechanisms linking ACEs and BC in adults. Results. From the 3321 papers found, nine articles involving 2931 participants were selected. All studies included ACEs retrospective assessments and psychological measures, and seven of them considered biomarkers. Individuals exposed to greater ACEs were at increased BC risk compared with individuals with no ACEs. Associations were found between child abuse and/or neglect, depression, perceived stress, fatigue, and plasma levels of cytokines interleukin (IL-6), C-reactive protein (CRP), soluble tumor necrosis factor receptor type II (sTNF-RII), interleukin IL-1 receptor antagonist (IL-1ra), and psycho-physiological adjustments that may lead to BC. Conclusions. Exposure to multiple ACEs appears a risk factor for BC development in adulthood. Although the clinical relevance of findings such as this is ambiguous, the review added evidence for a link between the presence of childhood adversity and BC occurrence, pointing to psychological, hormonal, and immunological dysregulations.

## 1. Introduction

Breast Cancer (BC) is one of the most diagnosed female cancers and a leading cause of death. Factors such as age, early menarche, familiarity, late menopause, use of contraceptives, lifestyle, and environmental factors are stated as BC risk factors. Some other research directions have explored the effect of distress in cancer occurrence with conflicting evidence. For example, a meta-analysis of cohort research stated that a history of stressful life events might somewhat enhance BC risk [1], while another [2] has detected no relation between adverse life events and BC incidence. These conflicting findings can be attributed to the differences in approach, control for confounding variables, populations, research designs, and measures of adverse life events.

Epigenetic changes are a field area of attention as they play a role in overexpression of oncogenes or silencing of tumor suppressor genes, therefore stimulating tumorigenic pathways in BC. Human development is managed by epigenetic mechanisms that help distinguish and record environmental information and model cellular and physiological functions throughout the lifespan. One such mechanism is DNA methylation (DNAm), which usually worsens the expression of a gene [3]. The epigenome, comprising DNAm, influences genes expression and offers a molecular context for how the genome is influenced by environmental experience. DNAm may be a pathway by which environmental exposure to adversities becomes biologically rooted, supplying adverse mental and physical health [3,4,5]. There is rising interest in the relationship between childhood adversity and telomere shortening (the protective cap at the tips of chromosomes) [6,7]. Shorter telomeres are linked with bigger risk for earlier mortality and psychiatric diseases, including depression [3,6,8]. Moreover, high inflammation can stimulate T-cell proliferation, leading to the shortening of telomeres. Meaningfully, a meta-analysis noticed a significant relationship between the number of ACEs and telomere length [9]. These modifications in DNA may underlie the associations between early life stress and health outcomes. One challenge for future research will be to reveal what factors are involved and how they interact with epigenetic mechanisms to boost susceptibility or resilience to the harmful impacts of childhood adversity. For example, individual variations in genetics, age, stressors, and duration may affect the extent to which early life adversity influences the genome and behavioral outcomes.

Individuals with Adverse Childhood Experiences (ACEs) are inclined to have more physical and mental health problems in adulthood. ACEs influence children directly or indirectly, through their living ecosystems, and include child abuse/neglect, poor parent/child relationships, conflict, low socioeconomic status or extreme poverty, and other challenges during their sensitive developmental stages. ACEs are related to several negative health conditions (poor mental health, substance use disorder, adverse health behaviors, chronic physical disease, and shortened life span). Stressors trigger disparate patterns of response from the body’s systems making individuals more vulnerable to disease in adulthood [3]. The sympathetic–adrenal–medullary (SAM) system and the hypothalamic–pituitary–adrenocortical (HPA) axis are independently or jointly activated in response to psychosocial stress depending on the source and type of stressor [10]. SAM activation produces the release of catecholamines that cooperate with the autonomic nervous system to particularly regulate the cardiovascular and immune systems, generating mood changes [11]. The activation of the HPA axis promotes the production of cortisol [12]. Physiological and biomolecular research is progressively uncovering how childhood experiences involving chronic stress led to fluctuations in the development of nervous, endocrine, and immune systems, ensuing in impaired emotional, cognitive, and social functioning [10,11,12,13,14], amplified allostatic load (i.e., chronic physiological harm), and adoption of health-damaging behaviors [15,16,17] (See Figure 1).

Larger emotional reactivity to daily-life stressors is linked with an amplified inflammatory response [3]. Childhood adversity can alter the vulnerability to diseases due to plasticity, i.e., the capability—probably mediated by epigenetic mechanisms—to grow in several variable ways based on the primary milieu [3,10]. Extensive epidemiological studies have linked childhood abuse to an increased occurrence of traumatic stress symptoms in women with BC [18,19], though the path through which child maltreatment increases vulnerability to symptoms of cancer-related traumatic stress is still not completely clear. 

Felitti et al. [20] reported significant rates of prevalence of childhood abuse in 9508 individuals who completed standardized medical evaluations. Moreover, a relevant quantity–response relationship between the number of childhood injuries and the probability of cancer diagnosis in adulthood is proposed [21,22], together with several possible mechanisms explaining the dysregulation of emotional, cognitive, and neurobiological systems observed in survivors of child abuse and maltreatment [23]. A study by Warner et al. [24] proposed four biomarkers overall associated with BC (adiponectin, C-peptide, high sensitivity C-reactive protein, and insulin-like growth factor-1) that seem related to ACEs with an increased risk of developing estrogen-receptor (ER) and, therefore, BC. Nelson et al. [25] described potential outcomes in adult caused by ACEs, linking them to mental and physical health outcomes during the life through epigenetic, neurodevelopmental disruption/reprogramming stress, and immune regulatory systems. Previous reviews have summarized evidence for long-term health effects of ACEs through maladaptive copings [3,20,25], but the specific mechanisms have not yet been unquestionably ascertained. 

To our knowledge, no attempt has been made to summarize data for the BC risks associated with multiple ACEs. Furthermore, the amounts of perceived stress resulting from adverse events can be different in different individuals because the key is the distress suffered. The reasons for this heterogeneity are linked to a complex interaction between genes and environment, which modifies the individual recovery trajectory [4]. Despite the potential gains of widespread screening for ACEs being recently questioned, reducing their negative effects might be significant: even a 10% decrease in prevalence and severity of chronic physical and mental health disorders associated with ACEs would have a general impact on public health and health care costs [26]. Screening for ACEs in children could have the additional value of avoiding future adverse experiences. 

The main aim of the present review is to summarize studies examining the means by which differences in psychological facets and/or biomarkers can modulate individual vulnerability, contributing to a behavioral and immunological profile during BC development. This present review extends the findings of existing systematic reviews by expanding the coverage of literature (up to 30 June 2022) and including more databases, to reflect a rising interest around childhood maltreatment in recent years. 

Some research questions have led to the planning of this review. Which mechanisms are involved in the development of BC when women suffered from ACEs? What relationship exists between childhood adversity and inflammatory markers? Does childhood trauma precipitate a chronic inflammatory state that increases biomarker levels (including CRP, IL-6, and TNF-alpha) in BC adults during or after treatment? The full process of study identification, inclusion, and exclusion of papers in our review is depicted in Figure 2. 

## 2. Materials and Methods

### 2.1. Literature Search Methodology

A systematic search was conducted according to PRISMA statement guidelines for systematic reviews [27,28,29], and it followed the recommendations of the Cochrane Collaboration for systematic reviews [30]. To ensure a thorough and systematic search of the literature, two methods were used to retrieve relevant articles. First, electronic database research was performed using the search line, in EBSCO, PubMed and Scopus databases. Since the rest of the databases (Google Scholar, Science Direct, Annual Reviews, Cambridge Journals, and PNAS) could only accept a few parameters, the search line and parameters were modified to include minimal acceptance requirements specifying some parameters like year of publication (between 2010 and 2022), “breast cancer” and “child abuse” or “neglect”. Furthermore, to find any additional published studies, we manually searched other resources for relevant investigations through the screening of the reference lists of review articles.

The keywords used for the searching scheme (in title or abstracts) were an association of Child* OR infant* OR toddler* OR baby* OR babies OR kid* OR minor* AND abuse* OR neglect* OR maltreat* OR mistreat* OR violence* OR violat* OR explotation* OR offense* OR perver* OR rape* OR assault* OR attack OR brutality OR cruel* OR disturb* OR negligence OR transgress* AND cancer* OR tumor* OR tumour* OR neoplasm* OR carcinoma* OR oncology* OR metastasis AND Stress OR ptsd OR anxiety OR “post-traumatic stress disorder” OR “post-traumatic stress disorder” OR “posttraumatic stress disorder” AND HPA OR “dna methylation” OR cytokines OR neuropeptides OR vulnerability OR inflammation OR neurotoxicity or gen* AND Breast.

### 2.2. Eligibility Criteria and Study Selection

Figure 2 schematically shows the selection process of the papers. Articles were eligible for this review if the studies were published in peer-reviewed journals and described the association between ACEs and BC incidence. In addition, the psychological data collection had to have been carried out concurrently with a confirmed BC diagnosis and accessed through self-report measures, questionnaires, clinical interviews, or clinical diagnoses. Study reports had to have contained information about the stage of cancer and treatment.

For our purposes, ACEs are defined as exposure to ≥1 of the following before the age of 18 years: abuse (physical, sexual, or psychological), neglect (failure to provide or supervise), household challenges (parental death, illness or incarceration, substance abuse, domestic violence, financial problems), and other types of adversity. Adult BC incidence was identified as having a diagnosis at the age of 18 years or older. 

Case reports and papers that were not original research, or papers not written in English, were excluded. We screened every title and abstract to determine eligibility based on the criteria: (1) investigation of mechanisms that may link early traumatic experiences and BC (like biomarkers of inflammation, stress, immunological or psychological measures); (2) adult subjects; and (3) original research published between the years 2010 and 2022. 

Two reviewers (OP, RTC) independently conducted the electronic searches identifying 3321 articles, which were then examined for eligibility. Another article was identified by searching the reference lists. If both reviewers deemed an article to be irrelevant, it was discarded. Inconsistencies in interpretation and disagreements were resolved through group of discussion and, if necessary, by consulting with the third author (DP). After removing the duplicates and a first screening, 78 full-text articles were scrutinized, and their titles and abstracts were evaluated. Reasons for excluding studies were documented. After analyzing the articles in more detail, sixty-four papers were excluded because they did not comply with one or more of the inclusion criteria. The remaining 14 were read to make sure they satisfied the inclusion criteria, but three of them lacked a measure for childhood trauma, and two did not specifically refer to BC. Upon closer examination, there was consensus on nine studies.

To extract relevant information, the studies were categorized according to the specific lines of research. A coding protocol, consisting of the following items, was used: (1) paper characteristics; (2) sample characteristics; (3) methodological features; (4) results; (5) ana-lyses employed; and, if possible, (6) reported associations between ACEs and BC outcomes based on adjusted analyses (i.e., risk, odds). Two authors (RTC and OP) independently extracted the data. Disagreements were resolved by arbitration (DP). To investigate the potential influence of socioeconomic disparities, we also retrieved information on the distribution of ACE and BC by measures of demographic and socioeconomic variables, when reported. Both reviewers completed abstraction forms for all papers and then compared their results. Studies were further grouped by type of ACE measures (i.e., individual vs. summary scores), and biomarkers. The nine articles analyzed included observational and controlled clinical trials.

## 3. Results

The main characteristics of the studies included in this systematic review are summarized in Table 1. Papers showed several methodological differences. Summary effect estimates were not calculated due to the studies’ heterogeneity. The oldest publication was published in 2010 [21]. Concerning the geographic context, studies were all conducted in the USA in hospital settings except for the most recent paper (2021) [31], which was conducted online. All studies included a retrospective measure of ACEs; they are mainly cross-sectional design [21,31,32,33,34], one used a prospective cohort design [35] collecting at baseline self-reported data on past ACEs and BC information prospectively, and two used a longitudinal prospective design [36,37,38]. Globally, in the present review, 2931 women, mainly Caucasian, with an average of 54.01 years were involved; only one study included a control group of 1000 women with an average age of 55.7 years. The findings, indicating that ACEs are related with psychopathology and immune dysregulation across many indices (including proinflammatory cytokine production and immune response to a tumor), are summarized below. Literature on how early life stress may alter epigenetic processes was then discussed.

The research by Crosswell et al. [32] was carried out at 12 months post-treatment for BC at stages 0-IIIA with the aim to explore associations with higher levels of inflammatory markers one year after BC treatment. The Risky Families questionnaire [20], including “Abuse”, “Neglect”, and “Chaotic home environment” subscales, was used. Inflammatory markers such as plasma levels of cytokines interleukin (IL-6), C-reactive protein (CRP), soluble tumor necrosis factor receptor type II (sTNF-RII), and interleukin IL-1 receptor antagonist (IL-1ra), were also incorporated. Participants were additionally assessed through the Beck Depression Inventory-II (BDI) [39] and the Perceived Stress Scale (PSS) [40]. The sample consisted of 152 women. The study determined that the overall score of adversity was associated with higher levels of the proinflammatory cytokine IL-6. However, disaggregating the child adversity subscales, it was shown that growing up in a chaotic/conflictual home environment was related with increased levels of IL-6 and sTNF-RII. Significant associations were found with age, body mass index (BMI), ethnicity, and number of alcoholic drinks per week. Physical and/or emotional abuse was positively related with concentrations of IL-6. This relationship was moderated by depression and perceived stress.

Han et al. [36] explored the relationship between childhood trauma and increased stress, fatigue, and inflammation across three intervals. ACEs were assessed through the Childhood Trauma Questionnaire (CTQ) [41], which includes Physical, Sexual, and Emotional abuse, and Physical and Emotional neglect subscales. The finding was dichotomized into CTQ+ if scores met moderate to severe childhood trauma cutoff value, or CTQ- if they did not. Multidimensional Fatigue Inventory (MFI) [42], Inventory of Depressive Symptomatology-Self Report (IDS-SR) [43], and Perceived Stress Scale (PSS) [40] were used. Cytokine interleukins IL-6, IL-1ra, and CRP, as well as the expression of genes involved in inflammatory responses, were also considered. Eight of the 20 subjects reported ACEs. Participants with CTQ+ constantly displayed greater fatigue, depression, and perceived stress than respondents with CTQ-. Marital status lowered the impact of trauma on fatigue scores, whereas marital status, cancer stage, and age reduced the significance of the relationship between childhood trauma and depression. A positive association emerged between CTQ severity scores and depression, as well as between CTQ severity scores and perceived stress. Associations between inflammatory markers and behavioral symptoms in CTQ+ and CTQ- patients exhibited a positive correlation between fatigue with both CRP and IL-6 in CTQ+ participants. A correlation was also observed between depression and CRP, IL-6, and IL-1ra in CTQ+ participants; perceived stress also links up with CRP and IL-6. Finally, CTQ+ patients revealed variations in gene transcripts related to inflammatory signaling including IL-22, IL-17, and C-C chemokine receptor 5 signaling in macrophages and T lymphocytes. Childhood abuse has been tied to prominent markers of inflammation, including CRP, IL-6, and TNF-alpha.

Witek-Janusek et al. [37] investigated the effects of ACEs on immune outcomes relevant to BC, such as Natural Killer Cells Activity (NKCA) and the pro-inflammatory cytokine IL-6. Forty women were assessed through the Perceived Stress Scale (PSS) [40], Center for Epidemiologic Studies, Depression scale (CES-D) [44], Multidimensional Fatigue Symptom Inventory Short Form (MFSI-SF) [45,46], and Quality of Life Index (QLI) [47]. Childhood adversity was stated through the Childhood Adversity Questionnaire (CTQ) [41]. Natural killer activity against tumor targets was estimated using peripheral blood mononuclear cells (PBMC); to determine circulating IL-6 in plasma, samples of the enzyme linked immunosorbent assay (ELISA) were used. The first evaluation (T1) occurred at least 2 weeks after surgery and subsequent evaluations were T2 at 5 ± 2 weeks, T3 at 9 ± 2 weeks, T4 at 15 ± 3 weeks, and T5 at 34 ± 3 weeks. Participants’ mean age was 55.6 years (SD = 9.4); 78% of them had cancer at stage 0 or I, and had undergone surgery (15%), surgery + radiation (60%), or surgery + radiation + hormonal therapy (25%). The surgery applied was 77% breast conserving and 23% mastectomy. Age was associated with stress and fatigue, reaching significance when entered in combination with ACEs. Childhood emotional neglect/abuse persisted as a noteworthy predictor of depression, perceived stress, fatigue, and quality of life; therefore, greater levels of childhood emotional neglect/abuse reflected greater levels of depressive symptoms, perceived stress, fatigue scores, and lower quality of life. This pattern remained unchanged over the nine-month period. Childhood physical abuse was negatively associated with the initial levels of depressive symptoms. Women who reported lower levels of physical neglect exhibited faster improvement in quality-of-life compared to women who reported greater physical neglect.

Conversely, at the first assessment, women with higher levels of childhood emotional neglect/abuse had lower NKCA, the degree of which grew over time, but the trajectory was not influenced by childhood emotional neglect/abuse. As a result, women with greater childhood adversities displayed lower NKCA during the nine-month period. Childhood physical neglect remained a significant predictor. Even though all women initially had similar levels of circulating IL-6, those with lower levels of physical neglect during childhood exhibited a quicker decrease of IL-6 throughout the study, while circulating IL-6 levels increased slightly for women who reported larger childhood physical neglect.

Kamen et al. [34] explored the effects of childhood trauma and cortisol levels on cognitive functioning in 56 women treated with acupuncture or sham/control. ACEs were evaluated through the Traumatic Events Survey (TES) [48], and cognitive functioning through the Functional Assessment of Cancer Therapy-Cognitive Version 3 (FACT-Cog) [49]. The Insomnia Severity Index Questionnaire (ISI) [50], the CESD [44], and the State Trait Anxiety Inventory (STAI-State) [51] were also applied, together with salivary cortisol assessment. Data from the TES survey were dichotomized into “Experienced trauma” and “No trauma experienced”. The most experienced traumatic events were witnessing a serious injury (19.6%, *n* = 11), physical (17.9%, *n* = 10), and sexual abuse (14.3%, *n* = 8). Findings showed that women with adversity had lower FACT-Cog overall, and that they had lower FACT-Cog subscale scores, showing between-group differences. Unexpectedly, depression and anxiety did not differ among groups. Self-reported cognitive functioning was negatively correlated with insomnia and anxiety. Childhood trauma was related to cognitive functioning in a multivariate model including childhood trauma, controlling for age, college education or higher, time since last chemotherapy, depression, anxiety, and insomnia. Anxiety and insomnia were also independently associated with cognitive functioning. Significant differences were found between groups in waking cortisol (T1), and in the cortisol slope between morning and evening (T1-T3). Among potential mediators, college education determined a difference between women with and without trauma.

Fagundes et al. [33] explored the link between childhood adversity and the expression of Epstein-Barr virus (EBV) and cytomegalovirus (CMV), analyzing whether this relationship could be observed beyond the psychological distress that women experienced after BC diagnosis and treatment. The sample was of 108 women (104 EBV seropositive and 56 CMV seropositive). Eligible women had completed treatment for stage 0 to stage IIIA BC within the past three years and were at least two months after treatments. Adversity measures were: (1) mother’s death; (2) father’s death; (3) severe parental/marital problems; (4) family member with mental illness; (5) family member alcohol abuse; and (6) lack of one close relationship with an adult. ACEs were asked about directly. Psychological measures were: (1) CES-D [40], (2) Charlson index [52] (19 conditions with a weight from 1 to 6 points) for comorbidities, and (3) the Pittsburgh Sleep Quality Index [53]. Blood samples were also taken to evaluate EBV IgG antibody titers and CMV IgG antibody titers. Although 11.1% of participants had undergone only a surgery, 40.7% of them suffered surgery, radiotherapy, and chemotherapy. From the sample, 40.1% did not experience any childhood adversity, while 28.7% experienced one ACE, 19.4% two, and 11.1% experienced ACEs in three or more instances. Correlations among demographic variables revealed that women with a greater number of adversities had fewer years of education and showed a poorer sleep quality than those with fewer adversities. Childhood adversities were related with depressive symptoms, but age, cancer stage, and time since treatment were not. Time since last treatment and ACEs were associated with EBV and CMV antibody titers. Depressive symptoms and education have been not related to EBV and CMV antibody titers, so they do not mediate the relationship between childhood adversities and antibody titers. 

Bower et al. [38] recruited a sample of women after diagnosis with early-stage BC, but before the onset of adjuvant therapy (radiation, chemotherapy, and/or endocrine therapy), testing the hypothesis that ACEs would be related to alterations in immune-related gene expression in monocytes. Childhood maltreatment and depressive symptoms were explored through the Childhood Trauma Questionnaire (CTQ) and Center for Epidemiologic Studies Depression Scale (CES-D). Blood samples were used for immune assessment where CD14+ monocytes were isolated for RNA extraction and gene expression analyses. Findings indicated bigger NF-kB-binding motifs within the promoters of up-regulated vs. down-regulated genes in women with BC who suffered from ACEs, matched to women with BC who did not experience ACEs, and evaluations of Type I interferon signaling likewise revealed improved activity in monocytes from women with ACEs.

McFarland et al. [35] investigated the association between ECAs and anxiety, depression, and distress in 125 patients with BC diagnosed within the previous five years and who were on stages 0-IV, had undergone mastectomy (50%) or lumpectomy (44.6%), had received chemo (55.6%), or were taking anti-hormonal therapy (52%). ACEs were assessed through the RFQ [20]. The Distress Thermometer and Problem List (DT and PL) [54] was used and anxiety and depression were also explored (HADS-A; HADS-D) [55]. No biomarkers were considered. The RFQ score was higher in participants who met criteria for distress, anxiety, and depression, and it was higher in women taking antidepressants, without difference due to ethnicity, civil status, or employment. The overall RFQ score was associated with depression and emotional problems. According to the authors, this picture was consistent with data showing elevated cytokine profiles in individuals who suffered ACEs and depression.

Goldsmith et al. [21] explored the relationship among childhood abuse and cancer-related intrusive and avoidant symptoms in 330 newly diagnosed individuals. ACEs were evaluated through the CTQ [32] and traumatic stress symptoms associated to BC over the previous two weeks through the Impact of Events Scale (IES) [56]. Twenty-five percent of participants were currently receiving medical treatment and 39% had undergone chemotherapy, 30% radiation, 15% a combination of radiation and chemotherapy, 8% surgery, and 11% other treatments. Age was negatively related to intrusive symptoms (e.g., unwanted cancer-related thoughts, emotions, images, or dreams). African Americans and women with lower incomes reported higher levels of avoidant symptoms (e.g., attempts to avoid cancer-related feelings, thoughts, or reminders). A multiple regression analysis showed that no abuse subscale was linked to avoidant symptoms, while intrusive symptoms were independently negatively associated to age and days from diagnosis, and positively related to emotional abuse. These data may reflect the cognitive and emotional schemas, hypervigilance, dysregulated stress responses, and altered neurological systems observed in survivors of childhood abuse [57,58,59,60]. A cancer diagnosis may trigger negative cognitions and emotions that are consistent with the patients’ prior traumatic experiences.

Wahbeh et al. [31] completed an anonymous cross-sectional study to explore the relationship between childhood (and adult) life events, emotional and psychological experiences, and BC status (time since diagnosis, age, method of BC discovery, breasts, lymph node involvement, treatment, current treatment if applicable, remission status, recurrence if applicable, last mammogram, and BRCA1 or BRCA2 status). They reached 2041 women through an online survey, 1041 were (or had been) BC patients and 1000 women were evaluated as controls. ACEs were estimated through four instruments: (1) CTQ (short form) also exploring the frequency of adverse experiences and the age of occurrence; (2) the item #7 from the Adverse Childhood Experience Questionnaire (ACEq), which explores whether the subject had seen her mother being physically abused as a child and as an adult; (3) Life Events Checklist (LEC), a self-report on potentially traumatic events in lifetime; two items concerning death of a loved one and divorce (or major break in relationships) were added with an additional set of questions addressing emotional, psychological, and energetic trauma (due to control, neglect, or abuse), not covered in the previous questionnaires; and (4) The Post Traumatic Growth Inventory Short Form (PTGI-SF) on the transformative growth from adverse experiences was employed. The participants were on Stage 0 for 16.6% of them, while 27.8% experienced Stage I, 27.1% Stage II, 15.9% Stage III, and 7.2% Stage IV. BC status was associated with increased age at first live birth, years of education, and income. African American women had greater odds of BC than other ethnicities and most important health problems and hormone replacement therapy. Mother’s use of diethylstilbestrol (a non-steroidal estrogen medication) doubled and tripled the odds of BC. As for ACEs, a competitive environment and severe human suffering showed a relationship between childhood occurrence (and adulthood), with a subsequent BC diagnosis. Considering ACEs suffered at different ages, emotional neglect (0–7 years), physical neglect (8–18 years), sexual abuse (0–7 years), fire or explosion (8–18 years), exposure to a toxic substance (19–90 years), assault with a weapon (19–90 years), severe human suffering as a child (8–18 years) and as an adult, and a competitive environment at all ages, were associated with increased odds of BC.

### 3.1. Types of ACEs

ACEs may vary considerably from each other; therefore, it is difficult to find a straightforward and unique way to determine which events to consider as adverse childhood events and how these should be measured. Every study yielded different issues and used different tools to assess such events. The CTQ was administered in five investigations [21,31,36,37,38], but the data were managed in different ways. Goldsmith et al. [21] and Wahbeh et al. [31] used the brief screening version [61], which assesses three subscales: “Physical abuse”, “Sexual abuse”, and “Emotional abuse”, with neglect included in the abuse subscale. The other two studies [36,37] employed the complete version with the subscales “Emotional abuse” and “Neglect”. Whereas Witek-Janusek et al. [37] applied test scores, Wahbeh et al. [31] registered the frequency of exposure at different age rates and Han et al. [36] preferred to dichotomize the data, assigning a positive or negative value to a variable called CTQ: if a subject displayed a score on any of the subscales representing moderate to severe trauma, she was considered having childhood trauma and a positive value was assigned to the variable (CTQ+), alternatively the variable was negative signed (CTQ-). 

The RFQ [20] was administered in two investigations [32,35]; it is a 13-item questionnaire which includes “Abuse”, “Neglect” and “Chaotic home environment” subscales concerning experiences having occurred between 5 and 15 years. Crosswell et al. [32] reported RFQ scores plus values for each subscale, whilst MacFarland et al. [35] included the RFQ overall score. Kamen et al. [34] applied the TES [48] that assesses 30 traumatic experiences occurring in both childhood and adulthood, 10 of which closely childhood traumas (e.g., natural disaster, life-threatening illness, witnessed someone being killed, tortured, or sexually assaulted). This tool also contained the number of instances and the perpetrator of the adversity, as well as the helplessness/fear suffered during the event. The childhood trauma exposure was calculated as a sum of the number of traumatic events and dichotomized as experienced any trauma vs. no trauma.

Fagundes et al. [33] did not use a standardized questionnaire, rather asking participants directly if they had suffered one of six traumatic events before the age of 17 years (death of mother or father, severe parental marital problems, direct family member with mental illness or abusing alcohol, and lack of at least one close relationship with adults). Believing that the effects of childhood adversity on mental health are additive in adulthood, they generated a variable reflecting the overall number of adversities [54]. Wahbeh et al. [31] also used item #7 on the Adverse Childhood Experience (ACE) questionnaire, which asks if the subject has seen her own mother being abused, and the Life Events Checklist (LEC) concerning potentially traumatic events during lifetime. It assesses exposure to 16 events that may result in PTSD or distress plus one item assessing any other stressful event not captured in the first 16 items. Finally, a 13 items ad hoc questionnaire assessing further life events not included in the former instruments addressing emotional, psychological, and energetic trauma (of controlling, neglectful, or abusive nature) was added.

### 3.2. Methods Used to Manage the Occurrence of ACEs

The different methods implemented in the studies to measure ACEs, or the amount of them, may include: (1) an individual variable, (2) a portion of an ACE summary score, or (3) both. Some studies employed ACEs data to create dichotomous variables, such that participants were scored positive or negative for their ACE exposure [36]. In other research [31], authors generated a summary score indicating the number of ACEs, considering only the summary score or the frequency of exposure to natural disasters and environmental accidents, violence, combat, imprisonment, severe or indirect suffering in childhood, and subsequently experiencing similar trauma in adulthood. In still others [27], only the relationship between an ACE summary score and a cancer outcome was analyzed; consequently, it remains unknown what type of ACE could have contributed to the increase of BC risk. Bower et al. [38] categorized participants into one of three groups based on ACEs type/severity (score 0: no maltreatment scored; score 0.5: physical and/or emotional abuse or neglect but no sexual abuse scored; score 1: sexual abuse with or without physical and/or emotional abuse or neglect). Finally, further studies made a distinction between cancer and specific kinds of ACEs, such as abuse, neglect, or chaotic home environment [32,35].

Many ACEs were considered. Emotional neglect and emotional abuse were the most frequent adversities scrutinized [21,35,37], followed by chaotic environment and neglect [32,35,37], emotional neglect [32,36], and sexual abuse [21]. In two studies [31,34], about 47% of participants reported ACEs without specifying the type. Witnessing a serious injury was included in one research [34]. The research of Fagundes et al. [33] does not characterize the ACEs experienced, reporting only that 64% of the respondents had childhood adversities. The same happens in the Bower et al. study where, based on ACEs exposure, 28% of the sample was classified as having suffered from physical and/or emotional abuse or neglect and 7% as experiencing sexual abuse [38].

### 3.3. Stress Vulnerability and Its Clinical Expressions

Most studies have explored to what extent an environmental experience (such as ACEs) shaped a potential phenotypic alteration. It was revealed that patients with BC who have undergone ACEs had psychosocial problems and a profile of biological reactions centered on higher expression of inflammatory biomarkers, such as the IL-6 [32]. This perspective focused on the relationship between ACEs as a first wave of severe stressors in the beginning of life, making the organism more sensitive when facing BC, which acted as a second wave of distress increasing the individual’s vulnerability [23]. 

All the studies incorporated in the review except two [20,31] involved depression and anxiety measures. Four also included a stress scale [32,35,36,37]. Furthermore, two studies contained a questionnaire exploring fatigue [36,37], two included an index of comorbidities [33,37], one the quality of life [37], one a functional assessment of cognitive therapy [35], one involved a measure of the impact of events [21] and, finally, one included a measure of resilience from which only items from the subscales “New possibilities” and “Appreciation of Life” are included [31].

### 3.4. Inflammation Markers

Three studies [32,36,37] contemplated inflammatory markers such as plasma levels of IL-6, and two the CRP [32,36]. One investigation [36] also included IL-1ra. Other inflammatory biomarkers incorporated were NKCA [37], sTNF-RII [62], gene expression [36,38], cortisol [34], and latent herpesviruses (EBV and CMV) [33]. Evidence emerged that childhood abuse is associated with elevated IL-6 [32,36] and cortisol [34,63,64]. Examining the relationship between inflammatory markers and RFQ subscales, findings suggested that risky factor total score was associated with IL-6 and marginally associated with CRP. In particular, the dimension of abuse was related with IL-6 while chaotic environment with IL-6 and TNF [32,37]. Childhood adversities were also correlated with EBV and CMV antibody titers [34], emphasizing a consequential dysregulation of the immunological response. In Bower et al. [38], genome-wide transcriptional profiling of isolated monocytes identified 202 gene transcripts that differed in average expression level by > 25% over the range of maltreatment exposure, with a considerably larger prevalence of NF-κB-binding motifs within the promoters of up-regulated vs. down-regulated genes in women exposed to ACEs, suggesting more inflammatory signaling. The analyses of Type I interferon signaling also showed higher prevalence of Interferon Response Factor (IRF)-related binding sites in individuals who suffered from ACEs. This finding persisted as significative where the analyses controlled for current depression. Nevertheless, NF-κB and IRF-related gene expression were greater in women with both ACEs and current depression.

## 4. Discussion

Evidence from population-based investigations has revealed that children who suffered trauma are about twice as likely to develop a mental disorder compared to those who have never experienced these experiences [65,66]. Among individuals with the same diagnosis, those who had ACEs exhibit a more severe clinical presentation of the disease, with less successful treatment outcomes [67]. Much of the research aimed to explore the relationship between childhood adversity, referred to as ACEs [20] or Early Childhood Adversities (ECAs) [35], and subsequent health outcomes. Even though ACEs definitions differ between studies, they usually include child maltreatment (e.g., child abuse and neglect) and other household adversities. 

A consistent association between ACEs and adverse clinical outcomes has been reported, suggesting that ACEs may alter the stress response inhibiting the optimal development and leading to more sensitive reactivity throughout the lifespan (see Figure 1). As a result, individuals who experienced greater adversity in early life may be at greater risk of late life health outcomes [62].

This emphasizes the value of identifying the vulnerability factors that contribute to individual differences in both the psychological response to adverse life events and adult health [68]. In this context, it is crucial to identify pathways between ACEs and cancer development in adulthood with their potential biological, physiologic, and behavioral mechanisms [69].

Evidence indicated that ACEs are associated with an increased risk of well-established cancer risk factors (e.g., tobacco or alcohol use), and chronic conditions may be part of the causal pathway from early adversity to cancer [1,2]. Studies among individuals with basal cell carcinoma, who had suffered from life stressors in the year preceding cancer development, indicated that patients with childhood maltreatment were more likely to display a minor immune response to the basal cell carcinoma onset. Lesser levels of messenger RNA (mRNA) coding for immune markers were shown [70]. 

This review focuses on ACEs and BC development and, extending the results found in previous reviews, showed that some ACEs are associated with an increased risk of BC [1,2]. It includes psychological variables, biomarkers, and BC status evaluating articles which explored the psychological factors of participants concurrently to a BC diagnosis. 

The relation between childhood adversity and BC is endorsed by some assumptions. According to one, the distress triggered by such events can influence HPA axis and produce disorders in the endocrine system, and thus the quantity of cortisol is raised, and anti-neoplastic activity is reduced, giving rise to an adult phenotype with a particular reactivity to stress [65,71,72]. Furthermore, studies highlighting the relationship between ACEs and inflammation in individuals with BC indicate that elevated inflammation is associated with BC recurrence [73,74]. The notion of neuroinflammatory sensitization postulates that ACEs can potentiate the stress response; inflammation, in turn, can signal the central nervous system and promote depressive symptoms through behavioral and neurobiological alterations. Adverse experiences can contribute to a protoinflammatory phenotype in people with depression, as showed by Danese et al. [75]. They indicated that individuals with current major depression, who suffered childhood maltreatment, were 1.48 times more likely to show high CRP levels than depressed individuals without childhood maltreatment. Synthetically, this finding highlights the construct of neuroinflammatory sensitization that binds stress, inflammation, and depression. Early stress enhances the risk of inflammation through several mechanisms, increasing sympathetic activity and decreasing parasympathetic activity with norepinephrine that endorses the production of proinflammatory cytokines, while parasympathetic activity works with the cholinergic anti-inflammatory way. A second process through which childhood stress can raise inflammation is via enhanced transcription of nuclear factor kappa B cells (NF-κB), an intracellular signaling molecule that controls proinflammatory cytokine gene expression. The increased transcription of NF-κB can happen through epigenetic processes [3]. Furthermore, Bower et al. [38], evaluating childhood maltreatment and monocyte gene expression among women with BC, suggested a greater inflammatory signaling among women with BC who had undergone childhood maltreatment.

Epigenetic pathways are very complex, and the recent sequencing of the entire exome of various cancers, including BC, has found that mutations in genes that regulate the epigenome are extremely pervasive [76,77]. Given that the epigenome functions are at the head of the pyramid of gene control mechanisms, this implies that the mutations in genes controlling the epigenome may affect BC through multiple pathways. Mechanisms other than DNA methylation and shortened telomeres can also assist in the regulation of stress pathways (e.g., histone modifications and noncoding RNA) [9,62,78,79]. DNA methylation, an epigenetic marker [80], could persist along adulthood making individuals susceptible to psychopathology, probably through HPA activity. Using a genome-wide sequencing approach to transcriptionally profile circulating monocytes, research [38] indicated greater NF-κB-binding motifs within the promoters of up-regulated vs. down-regulated genes in patients with BC who suffered from ACEs, compared to women with BC who had not experienced ACEs. This evidence is maintained controlling for depression. People exposed to childhood maltreatment or neglect had enrichment of variably methylated CpG sites (vCpGs) in genes regulated by hormonal receptors, including the glucocorticoid receptor (NR3C1) gene [81]. Similarly, among individuals with borderline personality disorder, exposure to trauma during childhood was linked with an increase in the methylation of the BDNF gene (implicated in many psychopathologies) into peripheral blood leukocyte cells. Indeed, the responders to the therapy showed a reduction in BDNF methylation [80,82]. 

The studies included in our review assessed ACEs retrospectively and DNAm mainly in adulthood. Therefore, it is not possible to determine whether DNAm differences are produced by ACEs or by later experiences [78,79,83]. On the contrary, conducting surveys in youth would offer a chance to observe epigenetic patterns that could be examined reasonably soon after facing childhood adversities. Yehuda et al. investigated how the offspring of Holocaust survivors showed differential methylation of a gene known to be highly stress-sensitive, the glucocorticoid receptor (NR3C1) gene [84]. Sumner et al. explored whether the presence or absence of ACEs was associated with DNAm, using epigenome-wide analyses in a sample of 113 youths aged 8–16 years with a two-year follow-up period. They found that ACEs are linked to several epigenetic markers through life, suggesting the need for research integrating genetic and epigenetic information [85]. 

Although the research is in progress, and the direction of methylation in various peripheral tissues needs additional investigation, these data suggest that a program can be efficacious in reshaping our epigenome [81]. Further investigation should clarify how various ACEs have as consequences epigenetic markers, and whether interventions may alter these markers. Like epigenetic regulation, resilience is a mechanism extending over a continuum and dependent on context. Therefore, the impact of various early preventive measures targeting family programs to provide a more positive early living environment should be explored.

The strengths of the present review include a comprehensive search performed across multiple databases on the association between ACEs and BC, including biomedical evidence to highlight any link between childhood adversity and BC, pointing out psychological, hormonal, and immunological dysregulations. A few limitations bound the generalizability of our results. This review is not a meta-analysis: only primary studies were analyzed and integrated qualitatively. Second, there is also a considerable variation in the tools used for assessment of ACEs. In each article, the variables studied were not explored by the same measures, and the associations found between the factors and BC were not studied considering potential cross-cultural differences. A third limitation was that studies were performed in high-income countries, and little is known about how ACEs predict BC in low-income, high-violence conditions, where adversities are prevalent across the life-course. Although evidence suggests a link between ACEs and BC risk, there is a need to explore this relationship in a rigorous manner through the meta-analytic approach. It would be crucial to define a standardized way to measure ACEs to compare data from different research. Studies enclosed here lacked detailed information on the various dimensions of the adverse experiences (e.g., age of onset or duration of ACEs) or about BC familiarity, except for Wahbeh et al. [31].

However, our results are of interest because ACEs are a potentially modifiable factor. Prevention policies capturing the causes of ACEs in childhood could offer opportunities for dealing with adversities rather than just their consequences. In the UN 2030 Agenda for Sustainable Development, nations have committed to meet 17 Sustainable Development Goal (SDGs), placing attention on early childhood development as a means of securing lifelong health, and many different agencies can contribute to the prevention of ACEs such as maternity services and schools. Nevertheless, the question of whether the occurrences of ACEs precede BC, and the extent of the psycho-physiological changes due to ACEs, remain central issues. Even if human studies offer insight into a relationship between the epigenome and the etiology of health adversity and BC in women subjected to ACEs, it is hard to propose conclusive statements regarding causality. Despite this, this review adds to the evidence of a link between experience of ACEs and BC development, highlighting the consequences of living in at-risk family environments. However, future investigation should concentrate on improving methodological accuracy with designs that allow stronger causal inference, assessing generalizability and persistence of any effect.

## 5. Conclusions

Our findings provide a comprehensive overview of forms of interpersonal trauma suffered during childhood by women with BC, and the ways in which adverse events affect health. The findings are quite interesting and suggestive of how the recognition of psychological status could influence treatment and prognosis of BC.

Unfortunately, retrospective self-reports are susceptible to recall bias. Moreover, most prospective studies did not collect data on timing and length of exposure. Genetic variations and environmental risks that may affect the relationship between ACEs and BC were mostly unassessed. 

Although non-conclusive, the findings from our review highlight the consequences of living in family environments at risk. The distinct forms of ACEs, as well as the occurrence and severity that might contribute to BC risk, are also underexplored. To better appreciate the processes underlying the relationship between ACEs and future BC diagnosis, further work should increase the dimensions and incidence of ACEs measurement. However, the recognition of the psychological status could influence treatment and prognosis of BC. Furthermore, associating psychological factors to the risk of BC could increase public awareness about the effect of mental health in physical health outcomes, leading to more successful prevention strategies.

## Figures and Tables

**Figure 1 ijerph-19-12615-f001:**
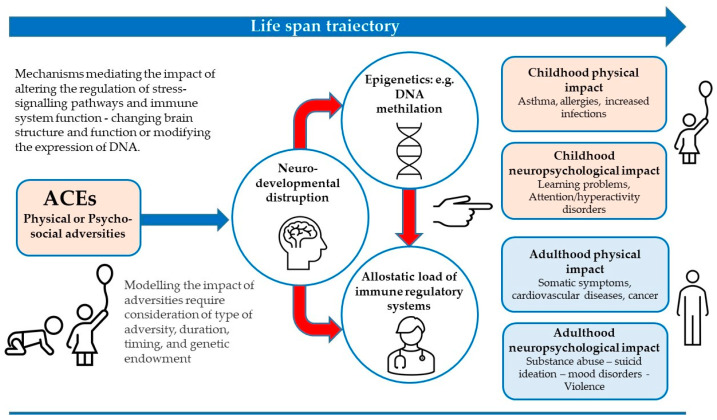
Impact of ACEs exposure and potential mechanisms for physical and/or neuro-psychological facets in childhood and adulthood.

**Figure 2 ijerph-19-12615-f002:**
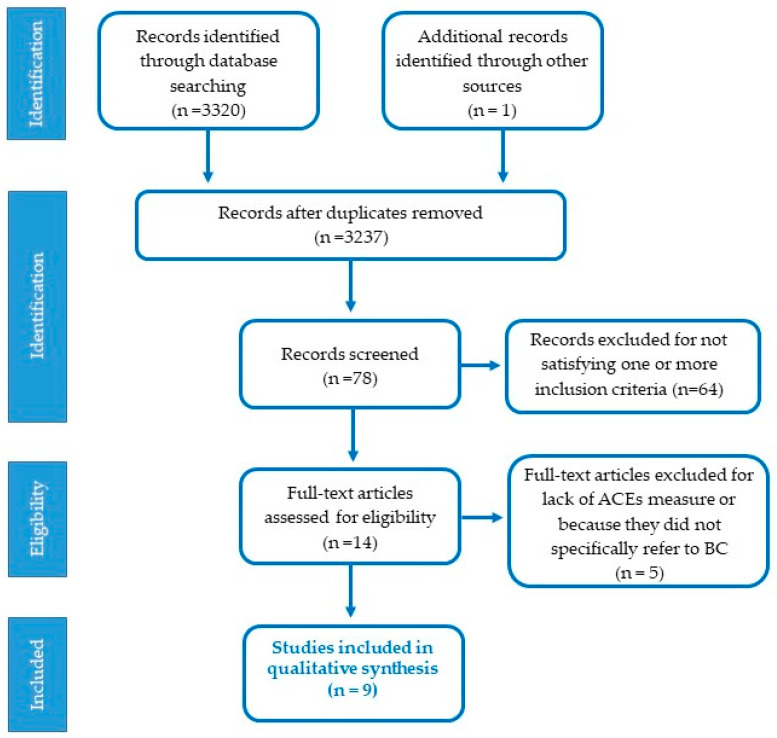
Selection process of the studies included in the paper.

**Table 1 ijerph-19-12615-t001:** Studies’ characteristics and association between ACEs and adverse clinical outcomes.

	Reference	Type of Survey	Sample Size	ACEs	Psychological Tools	Biomedical Measures	Associations	Stage
Inflammation biomarkers and psychological measures	Crosswell et al., 2014 [32]	Retrospective, cross-sectional, 1-year post-treatment	152 (21–65 years old) chosen from a wider cancer study; mean age = 51.7 (7.8)	Risky Family Questionnaire (RFQ) Score 27.75	(1) BDI ^1^ (2) PSS ^2^	IL-6 ^3^, CRP ^4^, IL-1ra ^5^, sTNF-RII ^6^	Inflammatory markers and RFQ ^7^ subscales. Risky factor total score with IL-6 ^3^ marginally associated with CRP ^4^; abuse and IL-6 ^3^; chaotic environment with IL-6 ^3^ and TNF ^8^.	0-IIIA
	Han et al., 2016 [36]	Retrospective-cohort, prospective longitudinal, 1-week pre-radiation; week 6 of radiation and 6 weeks after	20; mean age = 53.16 (11.6)	Childhood Trauma Questionnaire (CTQ)	(1) MFI ^9^ (2) IDS-SR ^10^ (3) PSS ^2^	Gene expression CRP ^4^, (IL)-6 ^3^, and IL-1ra ^5^	Childhood trauma and susceptibility of increased stress, fatigue, and inflammation during breast radiotherapy (RT).	0-IIIA
	Witek-Janusek et al., 2013 [37]	Cohort, retrospective, longitudinal prospective, 5 measures in 9 months	40 women (34 breast conserving surgery + radiation, 6 just surgery); mean age = 55.6 (9.4)	Childhood Trauma Questionnaire (CTQ)	(1) PSS ^2^ (2) CES-D ^11^ (3) MFI ^9^ (4) QLI ^12^	PBMC ^13^, NKCA ^14^, IL-6 ^3^	Childhood adversity factors vs. psychological measures and biomarkers over time.	Stages I, II and IIA (78% stage 0 or I)
Stress biomarker and psychological measures	Kamen et al., 2017 [34]	Cross sectional, retrospective; baseline, 3 weeks (mid-treatment), 6 weeks (end of treatment), 10 weeks follow-up	56 adult women diagnosed but not currently in treatment; mean of 53.6 years (9.8)	Traumatic Events Survey (TES)	(1) CES-D ^11^ (2) Charlson index (3) PSQI ^15^	Salivary cortisol	(1) ACEs vs. cognitive functioning, cancer treatment, time since treatment, depression, anxiety, and sleep.(2) Cognitive functioning vs. cortisol.	19.6% stage I 41.1% stage II 16.1% stage III 3.6% stage IV
Immunological and psychological measures	Fagundes et al., 2013 [33]	Cross-sectional, retrospective, from previous RCT	108 with EBV ^16^ (104) and/or CMV ^17^ (56) seropositive; mean age = 51.59 (9.39)	Six different conditions (e.g., death of mother or lack of close relationship with adult)	(1) CES-D ^11^ (2) Charlson index; (3) PSQI ^15^	EBV ^16^, CMV ^17^	(1) Relationship between ACEs and depressive symptoms. (2) Relationship between childhood adversities and EBV ^16^ and CMV ^17^ antibody titers.	0-IIIA
	Bower et al., 2020 [38]	Retrospective, longitudinal	86 with BS chosen from RISE study; mean age = 55.95 (11.8)	Childhood Trauma Questionnaire (CTQ) categorized in 3 group based on gravity (0–0.5–1)	(1) CES-D ^11^	CD14+ monocytes from PBMC, RNA extraction, gene expression analyses	(1) Relationship between ACEs and BC. (2) Relationship between ACEs, NF-κB-binding motifs, and depression. (3) Analyses of Type I interferon.	0-IIIA
	McFarland et al., 2016 [35]	Cohort, retrospective	125; mean age = 55.36 (13.20)	Risky Family Questionnaire: (1) Abuse; (2) Neglect; (3) Chaotic Home Environment	(1) DT a PL ^18^ (2) HADS ^19^-Anxiety; (3) HADS ^19^-Depression	None	ECA ^20^ vs. distress, anxiety, and depression.	0-IV within 5 years of diagnosis
	Goldsmith et al., 2010 [21]	Cross-sectional, retrospective	303 recruited from private and public hospitals; mean age = 50.68 (9.86)	Childhood Trauma Questionnaire (CTQ)	IES ^21^	None	Childhood abuse, particularly emotional abuse vs. levels of cancer-related intrusive and avoidant symptoms.	Not specified
	Wahbeh et al., 2021 [31]	Cross-sectional, retrospective Online surveys and cross-media measures	Clinical sample of 1041 women with BC (mean age = 57) Control group of 1000 women (mean age = 55.5)	(CTQ) short form + time (0–7; 8–18; 19–90 years old). Item #7 from Adverse Childhood Experience (ACE) questionnaire. Life Events Checklist (LEC). Ad hoc questionnaire.	Post Traumatic Growth Inventory Short-Form (PTGI-SF)	Basal metabolic index, reproductive health, hormone therapy, DES ^22^ history	Predictors of BC status: childhood trauma frequency (CTQ); trauma type; repetition across life; job experience.	16.6% (0) 27.8% (I) 27.1% (II) 15.9% (III) 7.2% (IV) 11.0% BRCA ^23^ +

^1^ BDI: Beck Depression Inventory; ^2^ PSS: Perceived Stress Scale; ^3^ IL-6: cytokines interleukin 6; ^4^ CRP: C-reactive protein; ^5^ IL-1ra: interleukin IL-1 receptor antagonist; ^6^ sTNF-RII: soluble tumor necrosis factor receptor type II; ^7^ RFQ: Risky Family Questionnaire; ^8^ TNF: tumor necrosis factor; ^9^ MFI: Multidimensional Fatigue Inventory; ^10^ IDS-SR: Inventory of Depressive Symptomatology-Self Report; ^11^ CES-D: Depression scale; ^12^ QLI: Quality of Life Index; ^13^ PBMC: Peripheral Blood Mononuclear Cells; ^14^ NKCA: Natural Killer Cells Activity; ^15^ PSQI: Pittsburgh Sleep Quality Index; ^16^ EBV: Epstein-Barr virus; ^17^ CMV: cytomegalovirus; ^18^ DT and PL: Distress Thermometer and Problem List; ^19^ HADS: Hospital Anxiety and Depression Scale; ^20^ ECA: Early Childhood Adversities; ^21^ IES: Impact of Events Scale; ^22^ DES: Diethylstilbestrol; ^23^ BRCA: Breast Cancer gene.

## Data Availability

The data that support the findings of this study are available on request from the corresponding author, [DP].

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
