# Peer review of "A Comprehensive Review on Multifaceted Mechanisms Involved in the Development of Breast Cancer Following Adverse Childhood Experiences (ACEs)"

_ijerph, 2022, doi:10.3390/ijerph191912615_

Round 1

Reviewer 1 Report

The authors carry out a literature review with the central aim of finding a potential connection between Adverse Childhood Experiences (ACEs) and breast cancer (BC) risk. The issue is interesting and as demonstrated by authors during the manuscript there is a significant number of papers that show the attention of the scientific community. However, it does not seem to me that the title should include the expression of potential mechanisms, given that the biochemical and/or pathophysiological evidences are scarce and speculative, which in my opinion only allows to present associations and some correlations at the epidemiological level. Likewise, the authors make several references to epigenetic mechanisms in particular of DNA methylation but do not present works in the literature that support this idea. In general the introduction is a bit long and with repeated ideas.I also suggest a more careful review of the role of inflammatory markers in breast cancer in particular and also better explaining some statements such as the constants between lines 509-511.

Author Response

Response to referee 1

The authors carry out a literature review with the central aim of finding a potential connection between Adverse Childhood Experiences (ACEs) and breast cancer (BC) risk. The issue is interesting and as demonstrated by authors during the manuscript there is a significant number of papers that show the attention of the scientific community. However, it does not seem to me that the title should include the expression of potential mechanisms, given that the biochemical and/or pathophysiological evidences are scarce and speculative, which in my opinion only allows to present associations and some correlations at the epidemiological level.

We appreciate your comments and feedback. Your notes are in bold, and our comments back are not. You will notice I kept track changes so you can see our edits. According to your suggestion, we have changed the title of the paper changing "Potential Mechanism" to “Multifaceted Mechanisms”.

Likewise, the authors make several references to epigenetic mechanisms in particular of DNA methylation but do not present works in the literature that support this idea.

Thank you for pointing this out. We have included more details including addressing each of the points you made that you can find on current lines 569-576. We have now added the most recent data about telomere shortening in lines 48-52 as evidence of epigenetic mechanisms.

References added:

  • Hanssen, L. M.; Schutte, N. S.; Malouff, J. M.; Epel, E. S. The relationship between childhood psychosocial stressor level and telomere length: a meta-analysis. Health Psychol. Res. 2017, 5, 6378. doi: 10.4081/hpr.2017.6378
  • Tyrka, A. R.; Parade, S. H.; Price, L. H.; Kao, H. T.; Porton, B.; Philip, N. S.; et al. Alterations of mitochondrial DNA copy number and telomere length with early adversity and psychopathology. Psychiatry 2016, 79, 78–86. doi: 10.1016/j.biopsych.2014.12.025
  • Bower, J. E.; Kuhlman, K. R.; Ganz, P. A.; Irwin, M. R.; Crespi, C. M.; Cole, S. W. Childhood maltreatment and monocyte gene expression among women with breast cancer. Brain Behav. Immun. 2020, 88, 396–402. doi: 10.1016/j. bbi.2020.04.001

In general, the introduction is a bit long and with repeated ideas.

We agree with the reviewer. As suggested, we have edited the introduction to make it clearer and more concise.

I also suggest a more careful review of the role of inflammatory markers in breast cancer in particular and better explaining some statements such as the constants between lines 509-511.

We thank the reviewer for pointing this out. We apologize for not addressing well the role of inflammatory markers. We have attended this point adding a further paper from Bower, et al. (2020). We thought it important to reference this paper, since it provided the most recent with respect to childhood maltreatment and monocyte gene expression among women with breast cancer. We direct the reviewer to the following frames on lines 504-513 where the details of the study are presented. To further clarify, we have now incorporated current lines 547-552 into discussion section.

Other changes:

  1. Other references added:
  • Page, M.J.; McKenzie, J.E.; Bossuyt, P.M.; Boutron, I.; Hoffmann, T.C.; Mulrow, C.D.; Shamseer, L.; Tetzlaff, J.M.; Akl, E.A. et al. The PRISMA 2020 statement: an updated guideline for reporting systematic reviews. BMJ 2021, 372 (71) doi: https://doi.org/10.1136/bmj.n71   
  • Danese, A.; Moffitt, T. E.; Pariante, C. M.; Ambler, A.; Poulton, R.; Caspi, A. Elevated inflammation levels in depressed adults with a history of childhood maltreatment.  Gen. Psychiatry2008, 65, 409–415. doi: 10.1001/archpsyc.65.4.409

  1. Abstract, Graphical Abstract, Figure 1, Figure 2 and Table 1 have been modified.

Reviewer 2 Report

Perfect review shedding the light onto very interesting topic

the review is very well-written, structured and to the point

I want to congratulate the authors for such publication

I have no major comments

I would only recommend some linguistic polishing to the text  and a more illustrative figure that would be more reader-friendly than the flow charts provided in the figure

Author Response

Response to referee 2

Perfect review shedding the light onto very interesting topic

the review is very well-written, structured and to the point

I want to congratulate the authors for such publication

I have no major comments

I would only recommend some linguistic polishing to the text and a more illustrative figure that would be more reader-friendly than the flow charts provided in the figure

We appreciate the reviewer’s estimation for our manuscript and her/his comments. We recognize that the manuscript needs for a linguistic polishing. Therefore, text has been reviewed for English. We acknowledge that that the flow-chart provided in the figure is challenging. We certainly agree that it will be important to include figure more reader friendly. Therefore, we decided to change the wording and framing text into the boxes to include a more specification about methodology and author decisions. However, the current paper is a systematic review paper: As such, the flow-chart is designed to help authors transparently report what the authors did, and what they found. We hope the included methods and procedures will help with the clarity.

Other changes:

  1. Othere references added:
  • Hanssen, L. M.; Schutte, N. S.; Malouff, J. M.; Epel, E. S. The relationship between childhood psychosocial stressor level and telomere length: a meta-analysis. Health Psychol. Res. 2017, 5, 6378. doi: 10.4081/hpr.2017.6378
  • Tyrka, A. R.; Parade, S. H.; Price, L. H.; Kao, H. T.; Porton, B.; Philip, N. S.; et al. Alterations of mitochondrial DNA copy number and telomere length with early adversity and psychopathology. Psychiatry 2016, 79, 78–86. doi: 10.1016/j.biopsych.2014.12.025
  • Bower, J. E.; Kuhlman, K. R.; Ganz, P. A.; Irwin, M. R.; Crespi, C. M.; Cole, S. W. Childhood maltreatment and monocyte gene expression among women with breast cancer. Brain Behav. Immun. 2020, 88, 396–402. doi: 10.1016/j. bbi.2020.04.001
  • Page, M.J.; McKenzie, J.E.; Bossuyt, P.M.; Boutron, I.; Hoffmann, T.C.; Mulrow, C.D.; Shamseer, L.; Tetzlaff, J.M.; Akl, E.A. et al. The PRISMA 2020 statement: an updated guideline for reporting systematic reviews. BMJ 2021, 372 (71) doi: https://doi.org/10.1136/bmj.n71   
  • Danese, A.; Moffitt, T. E.; Pariante, C. M.; Ambler, A.; Poulton, R.; Caspi, A. Elevated inflammation levels in depressed adults with a history of childhood maltreatment.  Gen. Psychiatry2008, 65, 409–415. doi: 10.1001/archpsyc.65.4.409

Abstract, Graphical Abstract, Figure 1, Figure 2 and Table 1 have been modified. Finally, we have edited the introduction to make it clearer and more concise.

We appreciate the Reviewers suggestions and hope that our revisions have satisfactorily addressed the Reviewers concerns.

Round 2

Reviewer 1 Report

Although the authors try to improve the aspects that I mentioned in the first review, I consider that they still do not satisfy. Epigenetic factors are much more than DNA methylation (eg, histone code) and just one of the factors (important, no doubt) in human development. The importance given to epigenetics in the manuscript do not have studies that consolidate the objective of the article. On the other hand, the inflammatory component (acute? chronic?) is a general mechanism of response to any aggression and considering the works already published on ACEs, the discussion component should be increased (here, more speculation is justified, that is, the critical view of the authors). My biggest concern is that despite being a work that deserves to be published, it is not clear that the review article demonstrates an association between ACEs and breast cancer or that it relates ACEs to the etiology of the disease.

Author Response

Response to referee 1

We appreciate your comments and feedback. Your notes are in bold, and our comments back are not. You will notice we have only kept track of new changes in the text, so you can see our last edits. 

We thank this reviewer for their careful analysis of our paper. We are glad the reviewer considers this a work that deserves to be published, and we believe the topic would be of interest to readers of the journal.

Although the authors try to improve the aspects that I mentioned in the first review, I consider that they still do not satisfy.

Epigenetic factors are much more than DNA methylation (eg, histone code) and just one of the factors (important, no doubt) in human development. The importance given to epigenetics in the manuscript do not have studies that consolidate the objective of the article.

We appreciate this comment and suggestion. We have reorganized and rewritten the discussion addressing this point more clearly, discussing other studies and expanding on the literature base. We acknowledge that mechanisms different from DNA methylation and shortened telomeres may also promote the regulation of stress pathways (e.g.: histone modifications and noncoding RNA).

This issue has been discussed in the following sentences:

Lines: 569-575

(…) Epigenetic pathways are very complex, and the recent sequencing of the entire exome of various cancers, including BC, has found that mutations in genes that regulate the epigenome are extremely pervasive [76-77]. Given that the epigenome functions are at the head of the pyramid of gene control mechanisms, this implies that the mutations in genes controlling the epigenome affect possibly BC through multiple pathways. Mechanisms other than DNA methylation and shortened telomeres, can also assist in the regulation of stress pathways (e.g. histone modifications and noncoding RNA). (…)

Lines: 582-588

(…) People exposed to child maltreatment or neglect had enrichment of variably methylated CpG sites (vCpGs) in genes regulated by hormonal receptors, including the glucocorticoid receptor (NR3C1) gene [81]. Similarly, among individuals with borderline personality disorder, exposure to trauma during childhood was linked with an increase in the methylation of the BDNF gene (implicated in many psychopathologies) into peripheral blood leukocyte cells. Indeed, the responders to the therapy showed a reduction in BDNF methylation (…)

Lines: 601-608

(…) Although the research is in progress and direction of methylation in various peripheral tissues needs additional investigation, these data suggest that a program can be efficacious in reshaping our epigenome [81]. Further investigation should clarify how various ACEs have as consequence epigenetic markers, and whether interventions may alter these markers. Like epigenetic regulation, resilience is a mechanism extending over a continuum and depending on context. Therefore, the impact of various early preventive measures targeting family programs to provide a more positive early living environment should be explored. (…)

In addition, we critically argued that the studies included in our review assessed ACEs retrospectively and DNAm mainly in adulthood and that with this evaluation cannot be ascertained whether DNAm differences are produced by ACEs or by later experiences. In this regard, we suggested that investigation performed on youths would offer a chance to observe epigenetic patterns that might be examined reasonably soon after facing childhood adversities. Therefore, the sentences below have been added:

Lines: 589-599

(…) The studies included in our review assessed ACEs retrospectively and DNAm mainly in adulthood. Therefore, it is not possible to determine whether DNAm differences are produced by ACEs or by later experiences [78-79, 83]. On the contrary, conducting surveys in youth would offer a chance to observe epigenetic patterns that could be examined reasonably soon after facing childhood adversities. Yehuda et al investigated how the offspring of Holocaust survivors showed differential methylation of a gene known to be highly stress-sensitive, the glucocorticoid receptor (NR3C1) gene [84]. Sumner et al., explored whether the presence or absence of ACEs was associated with DNAm using epigenome-wide analyses in a sample of 113 youths aged 8-16 years with a 2-year follow-up period. They found that ACEs are linked to several epigenetic markers thorough life, suggesting the need for research integrating genetic and epigenetic information. (…)

On the other hand, the inflammatory component (acute? chronic?) is a general mechanism of response to any aggression and considering the works already published on ACEs, the discussion component should be increased (here, more speculation is justified, that is, the critical view of the authors).

Thank you for pointing this out. As noted above, we have rewritten the discussion and clarified the role of inflammation. We have added more details about this point:

Line 548-553

(…) Furthermore, studies highlighting the relationship between ACEs and inflammation in individuals with BC indicate that elevated inflammation is associated with BC recurrence [73, 74]. The notion of neuroinflammatory sensitization postulates that ACEs can potentiate the stress response; inflammation, in turn, can signal the central nervous system and promote depressive symptoms through behavioral and neurobiological alterations.(…)

We have acknowledged that inflammation is a general mechanism of response and we have clarified the notion of neuroinflammatory sensitization and how adverse experiences can contribute to a proto-inflammatory phenotype also citing Bower et al study evaluating childhood maltreatment and monocyte gene expression among women with BC. We believe the conceptual foundation is now clearer.

My biggest concern is that despite being a work that deserves to be published, it is not clear that the review article demonstrates an association between ACEs and breast cancer or that it relates ACEs to the etiology of the disease.

Line 548-553

We appreciate the reviewer’s note and acknowledge that the discussion about the relationship between ACEs and breast cancer was not clear. We agree with the reviewer that caution is warranted when suggesting a link with the etiology of the disease. We have restructured the discussion to accommodate these new comments. Studies highlighting the relationship between ACEs and inflammation among BC individuals indicating that elevated inflammation is associated with BC recurrence are added (Bower, 2007; Danese et al.; Tabassum and Polyak, 2015).

Discussion: Line 633-637

(…) Even if human studies offer insight into a relationship between the epigenome and the etiology of health adversity and BC in women subjected to ACEs, it is hard to propose conclusive statements regarding causality. Despite this, the review adds to the evidence of the link between presence of ACEs and BC development highlighting the consequences of living in at-risky family environments. However, (…)

As there are no comments with regards other sections, we have not changed these sections of the manuscript except for discussing the paper (as noted above) and elaborating the limitations section.

We have also rewritten the abstract to align with changes in the discussion section.

The reference list has been updated.

We hope that the current version of our manuscript in responses to the reviewer comments and our revision sufficiently addresses her/his concerns.

References added:

  • Marzi, S.J.; Sugden, K. Arseneault, L.; Belsky, D.W.; Burrage, J.; Corcoran, D.L.; Danese, A.; Fisher, H.L.; Hannon, E.; Moffitt, T.E.; Odgers, C.L.; Pariante, C.; Poulton, R.; Williams, B.S.; Wong, C.C.Y.; Mill, J.; Caspi, A. Analysis of DNA methylation in young people: limited evidence for an association between victimization stress and epigenetic variation in blood. Am J Psychiatry 2018, 175, 517-29. https://doi.org/10.1176/appi.ajp.2017.17060693
  • Baldwin, J.R.; Reuben, A.; Newbury, J.B.; Danese, A. Agreement between prospective and retrospective measures of childhood maltreatment: a systematic review and meta-analysis. JAMA Psychiat. 2019; 76, 584-93. doi:10.1001/jamapsychiatry.2019.0097
  • Martins, J.; Czamara, D.; Sauer, S.; Rex-Haffner, M.; Dittrich, K.; Dörr, P.; de Punder, K.; Overfeld, J.; Knop, A.; Dammering, F.; Entringer, S.; Winter, S.M.; Buss, C.; Heim, C.; Binder, E.B. Childhood adversity correlates with stable changes in DNA methylation trajectories in children and converges with epigenetic signatures of prenatal stress. Neurobiol Stress. 2021, 15:100336. doi: 10.1016/j.ynstr.2021.100336
  • Sumner, J.A.; Gambazza, S.; Gao, X.; Baccarelli, A.A.; Uddin, M.; McLaughlin, K.A. Epigenetics of early-life adversity in youth: cross-sectional and longitudinal associations. Clin Epigenet 2022, 14, 48. https://doi.org/10.1186/s13148-022-01269-9
  • O’Donnell, K. J.; Chen, L.; Macisaac, J. L.; Mcewen, L. M.; Nguyen, T.; Beckmann, K., et al. DNA methylome variation in a perinatal nurse-visitation program that reduces child maltreatment: a 27-year follow-up. Translational Psychiatry 2018, 8(1), 15. https://doi.org/10.1007/s42844-020-00015-5
  • Hanssen, L. M.; Schutte, N. S.; Malouff, J. M.; Epel, E. S. The relationship between childhood psychosocial stressor level and telomere length: a meta-analysis. Health Psychol. Res. 2017, 5, 6378. doi: 10.4081/hpr.2017.6378
  • Burns, S. B.; Almeida, D.; Turecki, G. The epigenetics of early life adversity: current limitations and possible solutions. Mol. Biol. Transl. Sci. 2018, 157, 343-425. doi: 10.1016/bs.pmbts.2018.01.008
  • Krause, B. J.; Artigas, R.; Sciolla, A. F.; and Hamilton, J. Epigenetic mechanisms activated by childhood adversity. Epigenomics 2020, 12, 1239–1255. doi: 10.2217/epi-2020-0042
  • Lindqvist, D.; Epel, E. S.; Mellon, S. H.; Penninx, B. W.; Révész, D.; Verhoeven, J. E. et al.. Psychiatric disorders and leukocyte telomere length: underlying mechanisms linking mental illness with cellular aging. Biobehav. Rev. 2015, 55, 333–364. doi: 10.1016/j.neubiorev.2015.05.007
  • Joshi, S.; Garlapati, C.; Aneja, R. Epigenetic Determinants of Racial Disparity in Breast Cancer: Looking beyond Genetic Alterations. Cancers (Basel) 2022; 14(8):1903. doi:10.3390/cancers14081903
  • Perroud, N.; Salzmann, A.; Prada, P.; Nicastro, R.; Hoeppli, M.E.; Furrer, S.; Ardu, S.; Krejci, I.; Karege, F.; Malafosse, A. Response to psychotherapy in borderline personality disorder and methylation status of the BDNF gene. Transl Psychiatry 2013, 3(1):e207. doi: 10.1038/tp.2012.140.
  • Yehuda, R.; Daskalakis, N.P.; Lehrner, A.; Desarnaud, F.; Bader, H.N.; Makotkine, I.; Flory, J.D.; Meaney, M.J. Influences of maternal and paternal PTSD on epigenetic regulation of the glucocorticoid receptor gene in Holocaust survivor offspring. Am J Psychiatry 2014, 171(8):872-880. doi: 10.1176/appi.ajp.2014.13121571
  • Bower, J. E. Cancer-related fatigue: links with inflammation in cancer patients and survivors. Brain Behav. Immun. 2007, 21, 863–871. doi: 10.1016/j.bbi.2007.03.013
  • Tabassum, D.P.; Polyak, K. Tumorigenesis: it takes a village. Nat Rev Cancer. 2015; 15(8), 473-483. doi:10.1038/nrc3971